# *AP001056.1*, A Prognosis-Related Enhancer RNA in Squamous Cell Carcinoma of the Head and Neck

**DOI:** 10.3390/cancers11030347

**Published:** 2019-03-11

**Authors:** Xiaolian Gu, Lixiao Wang, Linda Boldrup, Philip J. Coates, Robin Fahraeus, Nicola Sgaramella, Torben Wilms, Karin Nylander

**Affiliations:** 1Department of Medical Biosciences/Pathology, Umeå University, 90185 Umeå, Sweden; lixiao.wang@umu.se (L.W.); linda.boldrup@umu.se (L.B.); robin.fahraeus@inserm.fr (R.F.); sgaramellanicola12@gmail.com (N.S.); karin.nylander@umu.se (K.N.); 2RECAMO, Masaryk Memorial Cancer Institute, 656 53 Brno, Czech Republic; philip.coates@mou.cz; 3Équipe Labellisée Ligue Contre le Cancer, INSERM UMRS1162, Institut de Génétique Moléculaire, Université Paris 7, IUH Hôpital St. Louis, 75010 Paris, France; 4Department of Clinical Sciences/ENT, Umeå University, 90185 Umeå, Sweden; Torben.Wilms@vll.se

**Keywords:** *AP001056.1*, lncRNA, enhancer, SCCHN, *ICOSLG*, tumor immunity

## Abstract

A growing number of long non-coding RNAs (lncRNAs) have been linked to squamous cell carcinoma of the head and neck (SCCHN). A subclass of lncRNAs, termed enhancer RNAs (eRNAs), are derived from enhancer regions and could contribute to enhancer function. In this study, we developed an integrated data analysis approach to identify key eRNAs in SCCHN. Tissue-specific enhancer-derived RNAs and their regulated genes previously predicted using the computational pipeline PreSTIGE, were considered as putative eRNA-target pairs. The interactive web servers, TANRIC (the Atlas of Noncoding RNAs in Cancer) and cBioPortal, were used to explore the RNA levels and clinical data from the Cancer Genome Atlas (TCGA) project. Requiring that key eRNAs should show significant associations with overall survival (Kaplan–Meier log-rank test, *p* < 0.05) and the predicted target (correlation coefficient *r* > 0.4, *p* < 0.001), we identified five key eRNA candidates. The most significant survival-associated eRNA was *AP001056.1* with *ICOSLG* encoding an immune checkpoint protein as its regulated target. Another 1640 genes also showed significant correlation with *AP001056.1* (*r* > 0.4, *p* < 0.001), with the “immune system process” being the most significantly enriched biological process (adjusted *p* < 0.001). Our results suggest that *AP001056.1* is a key immune-related eRNA in SCCHN with a positive impact on clinical outcome.

## 1. Introduction

Squamous cell carcinoma of the head and neck (SCCHN) is the sixth most common cancer worldwide with an estimated 600,000 new cases and 300,000 deaths annually [1]. SCCHN comprises a group of cancers located in the oral cavity, nasal cavity, larynx, hypopharynx and oropharynx. Tobacco and alcohol consumption, and human papillomavirus (HPV) infection are the most common predisposing factors for SCCHN [2,3].

Evidence is accumulating that long non-coding RNAs (lncRNAs) are linked to human diseases [4,5]. In contrast with the classic notion that non-protein-coding transcripts are non-functional and thus regarded as junk RNA, lncRNAs are biologically active and carry out diverse functions such as transcriptional regulation in cis or trans, the organization of nuclear domains and post-transcriptional regulation of proteins or RNA molecules [6,7]. Next-generation sequencing has identified tens of thousands of lncRNAs from single-cell eukaryotes to humans [8]. LncRNAs are expressed in a tissue and cell-type specific manner, making them potential cancer biomarkers [4] and many SCCHN-related lncRNAs have been reported based on differential expression at a given significance level [9,10], even if the function of these disease-associated lncRNAs is largely unknown.

Enhancer RNAs (eRNA) are a subclass of lncRNAs transcribed within gene enhancers, a major type of cis-regulatory elements in the genome [11]. Potential models suggest that eRNAs interact with RNA polymerase II and transcription factors to facilitate promoter–enhancer looping and a consequent increase in the transcription of the corresponding downstream gene [12]. RNA synthesis from enhancers has been confirmed in the majority of human cells and tissues, and transcriptional regulation via eRNAs plays a role in cancer [13,14]. It has been shown in basal cell carcinoma that mutations in the eRNA elements of *ACTRT1* could impair enhancer activity and *ACTRT1* expression, leading to aberrant activation of Hedgehog signaling and the contribution to tumor development [15]. To the best of our knowledge, there are no reports on SCCHN-associated eRNAs [16].

Here we set out to identify prognostic eRNAs and their target genes in SCCHN. We found that the functionally unannotated long non-coding RNA *AP001056.1*, located within the tissue-specific enhancer of an immune checkpoint gene, *ICOSLG*, was significantly associated with survival in patients with SCCHN. High *AP001056.1* levels in SCCHN associated with elevated expression of *ICOSLG* and other immune genes, suggesting that *AP001056.1* is an immune-related eRNA with a positive impact on clinical outcome.

## 2. Results

### 2.1. Putative Prognostic eRNAs in SCCHN

Using the PreSTIGE algorithm, a total of 2695 ENCODE (Encyclopedia of DNA Elements database [17]) annotated lncRNA transcripts that are expressed from active tissue-specific enhancers, as well as 2303 predicted target genes, have previously been identified [18]. This transcript dataset was utilized for identifying putative eRNA-target pairs. To facilitate data exploration in TANRIC, a gene-based interactive web platform for exploring lncRNAs in cancer, we used Ensembl BioMart for conversion of transcript ID to gene ID. After that, the 2695 putative eRNA transcripts were mapped to their corresponding 1288 genes. Finally, according to RNA-sequencing data from 426 TCGA SCCHN patients provided by the TANRIC database, we identified 18 of the 1288 putative eRNA genes with levels significantly associated with overall survival (Table 1, Kaplan–Meier log-rank test, *p* < 0.05). When correlating levels of these 18 eRNAs with levels of their predicted target gene mRNAs, significant correlations were seen only for five (Spearman’s rank correlation coefficient *r* > 0.4, *p* < 0.001; Table 1).

### 2.2. LncRNA AP001056.1 is a Key eRNA in SCCHN

LncRNA *AP001056.1* was found to be the top prognostic putative eRNA showing a positive correlation between gene expression and levels of its predicted target *ICOSLG* (inducible T cell costimulator ligand). In SCCHN patients, the *AP001056.1*-high group (cut off value not provided by TANRIC) showed better overall survival compared to the *AP001056.1*-low group (Figure 1a, Kaplan–Meier log-rank test, *p* = 0.002). Additionally, *AP001056.1* and *ICOSLG* mRNA levels correlated with each other (Figure 1b, Spearman’s *r* = 0.504, *p* < 0.001). The prognostic effect of *AP001056.1* and its correlation with *ICOSLG* mRNA levels in other TCGA cancer types was investigated using TANRIC. As summarized in Table 2, the impact of *AP001056.1* on overall survival and *ICOSLG* was specific for SCCHN and glioblastoma only. 

### 2.3. AP001056.1 Expression is Subsite Specific and Associates with HPV Status

We further investigated *AP001056.1* transcript levels in different SCCHN subsites and found significant differences (Figure 2a, Kruskal–Wallis H analysis, *p* = 0.014), with the highest median level seen in tonsillar cancer (31 patients).

TCGA data were available for HPV in situ hybridization from 89 SCCHN patients (67 HPV-negative and 22 HPV-positive). Of these, *AP001056.1* and *ICOSLG* mRNA levels were available for 12 HPV-positive and 60 HPV-negative patients. The majority of HPV-positive tumors were from the tonsil (9 out of 12). The HPV-positive tumors showed higher levels of *AP001056.1* compared to HPV-negative tumors (*p* = 0.001, Figure 2b). Similarly, *ICOSLG* mRNA levels were also higher in HPV-positive tumors (*p* = 0.028, Figure 2c). To further evaluate the function of *AP001056.1*, we used the TANRIC tool to identify significantly co-expressed genes in SCCHN. A total of 1641 transcripts showed a significant correlation with *AP001056.1* (*p* < 0.001), including *ICOSLG*. Gorilla functional enrichment analysis revealed that the most significantly enriched biological process was a so-called “immune system process” (adjusted *p* < 0.001). A list of the immune genes including Spearman correlation coefficients ≥0.6 are shown in Table 3.

### 2.4. Validation of AP001056.1 and ICOSLG Levels by Reverse Transcription Quantitative PCR (RT-qPCR)

RT-qPCR was performed to investigate gene expression and the correlation between *AP001056.1* and *ICOSLG* in matched tumor-free and tumor samples from 12 patients with squamous cell carcinoma of the oral tongue (SCCOT), the most common head and neck cancer subtype [19]. Up-regulation of *AP001056.1* and down-regulation of *ICOSLG* was seen in tumors compared to tumor-free samples (*p* = 0.002, Figure 3a,b), and a significantly positive correlation between these two RNAs was seen in the tumors (Spearman’s rank correlation coefficient *r* = 0.867, *p* < 0.001) but not in the tumor-free controls (Figure 3c).

## 3. Discussion

eRNAs are a specific subclass of lncRNAs derived within gene enhancer regions and are able to act in cis to influence transcription of the corresponding gene. To improve our understanding of eRNAs in SCCHN, we set out to identify prognosis-related eRNAs in SCCHN using the following approach, we: (1) Searched for lncRNAs expressed from active tissue-specific enhancers as putative eRNAs, (2) identified a subset of these putative eRNAs that significantly correlated with overall survival and (3) from these, identified a further subset that correlated with their potential target. This novel approach enriches for putative eRNAs influencing SCCHN pathobiology and can be employed for other human diseases with sufficient RNA-sequencing data available.

According to our criteria, the functionally unannotated lncRNA *AP001056.1* was identified as the top key eRNA candidate in SCCHN. *AP001056.1* is located within the enhancer region of the *ICOSLG* gene. Its encoded protein, ICOSLG, is a ligand for the T-cell-specific cell surface receptor ICOS and acts as a costimulatory signal for T-cell proliferation and cytokine secretion [20]. Strong correlations between expression of *AP001056.1* and *ICOSLG* were seen in several types of cancer, whereas, the most significant impact on overall survival was seen in SCCHN, making it an interesting lncRNA for this tumor type.

A recent study showed evidence for an association between HPV and expression of *ICOSLG* in SCCHN [21]. We found *AP001056.1* to be present at higher levels in HPV-positive compared to HPV-negative SCCHN, and when comparing *AP001056.1* levels between tumors from different subsites, the highest level was seen in tumors from the tonsil, probably due to the fact that the majority of tonsillar cancer are HPV-positive [22]. These results once again emphasize the necessity to take the subsite into consideration when interpreting data from analysis of the group of SCCHN tumors.

As *AP001056.1* is a functionally unannotated transcript, we tried to clarify its role by identifying other co-expressed genes in addition to its predicted target *ICOSLG*. Surprisingly, besides *ICOSLG*, we found a total of 1640 genes with significant expression correlations with *AP001056.1*. Although eRNA functions are primarily performed in cis, several observations suggest that eRNA could mediate the expression of other genes in trans [7,12]. *AP001056.1* could thus be speculated to have direct or indirect trans effects. Still, correlations between transcripts do not necessarily imply causal relationships, and whether *AP001056.1* is a functional component of enhancer activity remains to be determined. Nevertheless, according to gene ontology enrichment analysis, the gene transcripts that correlated to *AP001056.1* are mainly involved in immune system processes. Together with the correlation with the immune checkpoint protein ICOSLG, these results indicate that *AP001056.1* has a role in regulating the immune response in SCCHN. Considering the disease-specific impact on patient survival, novel therapeutic approaches that could increase *AP001056.1* expression might help to induce protective immunity for effective treatment of these cancers.

## 4. Materials and Methods

### 4.1. Identifying Prognostic eRNAs in SCCHN through Integrated Data Analysis

A list of lncRNAs expressed from active tissue-specific enhancers and their assigned targets predicted by PreSTIGE [18,23] was obtained. Ensembl BioMart was used to obtain the mapping between Ensembl transcript ID and gene symbol. The levels of putative eRNAs and their clinical relevance in the SCCHN cohort in TCGA were investigated using the interactive web server TANRIC (the Atlas of Noncoding RNAs in Cancer) [24]. The TANRIC co-expression data were also studied to evaluate correlations between putative eRNA levels and their predicted target genes. Putative eRNAs with significant correlations with both overall survival (*p* < 0.05) and levels of their target genes (*p* < 0.001) were considered candidate key eRNAs in SCCHN. Next, the TANRIC lncRNA and corresponding target gene expression levels were studied across 20 cancer types to evaluate the differences related to cancer type. Finally, we downloaded TCGA SCCHN RNA-sequencing and clinical data using cBioPortal [25] to investigate differences in gene expression related to SCCHN tumor subsite and HPV status.

### 4.2. Gene Ontology Enrichment Analysis

Besides the predicted targets, other transcripts with significant correlation with the identified prognostic eRNAs were obtained through TANRIC. To investigate the possible functional characteristics of the eRNA related coding genes, gene ontology enrichment analysis was performed (identification of gene ontology terms that are significantly overrepresented in a given set of genes [26]) using the web-based tool Gorilla [27]. The enriched gene ontology terms with an adjusted *p*-value < 0.001 were considered as relevant biological processes.

### 4.3. Quantification of AP001056.1 and ICOSLG Transcript Levels in Tongue Cancer Samples

Reverse transcription quantitative PCR (RT-qPCR) was used to confirm the RNA-sequencing data. The transcript ID is ENST00000411956.1 (for gene *AP001056.1*) and ENST00000407780.7 (for gene *ICOSLG*). The isolated total RNA was sourced from matched tumor and tumor-free samples from twelve patients diagnosed with squamous cell carcinoma of the oral tongue (SCCOT) [28,29]. The study was approved by the Regional Ethics Review Board, Umeå, Sweden (Dnr 03-201 and Dnr 08-003 M) and performed in accordance with the Declaration of Helsinki. The written informed consent was obtained from all patients and healthy individuals. The cDNA was synthesized from 500 ng total RNA using the RevertAid H minus first strand cDNA synthesis kit (Thermo Fisher Scientific, Waltham, MA, USA). RT-qPCR was performed using an IQ5 multicolor real-time PCR detection system with IQ SYBR Green Supermix (Bio-Rad Laboratories, Hercules, CA, USA). Custom primers for *AP001056.1* (forward: ATGGCCAGGCTGATTTCGAA, reverse: TCCAGATGAAAATGCCGGCT) and *ICOSLG* (forward: TCTGCAGCAGAACCTGACTG, reverse: TTTCTCGCCGGTACTGACTG) were obtained from Thermo Fisher Scientific. Each sample was measured in triplicate and the results were normalized to the housekeeping gene *RPL13A* (forward: GTACGCTGTGAAGGCATCAA, reverse: GTTGGTGTTCATCCGCTTG) and *GAPDH* (sequences of primers not provided, Primerdesign Ltd, Southampton, United Kingdom).

### 4.4. Statistics

To compare gene expression among different tumor subsites, the non-parametric Kruskal–Wallis H test was performed and the non-parametric Wilcoxon signed-rank test was used to compare between matched tumor and tumor-free samples. Spearman’s rank correlation coefficient was calculated to evaluate correlation strength. All statistical tests were conducted in IBM SPSS Statistics 25. A two-sided *p*-value < 0.05 was considered statistically significant.

## 5. Conclusions

To conclude, we developed and applied a novel approach to identify key eRNAs in SCCHN. Our data suggest that lncRNA *AP001056.1* is a prognosis-related gene for SCCHN, functioning as a tissue-specific eRNA of *ICOSLG*. With a potential role in immune response, *AP001056.1* is a promising therapeutic target for patients with this tumor type when taking subsite into consideration.

## Figures and Tables

**Figure 1 cancers-11-00347-f001:**
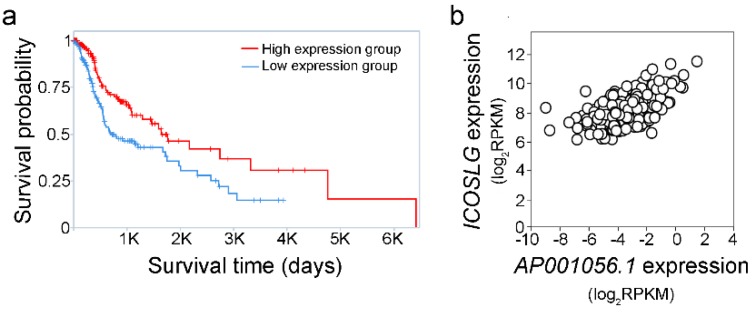
Impact of lncRNA *AP001056.1* on squamous cell carcinoma of the head and neck (SCCHN). (**a**) Kaplan–Meier overall survival curve for patients with *AP001056.1*-high and *AP001056.1*-low expression, obtained through the TANRIC (the Atlas of Noncoding RNAs in Cancer) platform. (**b**) Scatterplot showing the association between *AP001056.1* and *ICOSLG* levels according to TANRIC correlation analysis.

**Figure 2 cancers-11-00347-f002:**
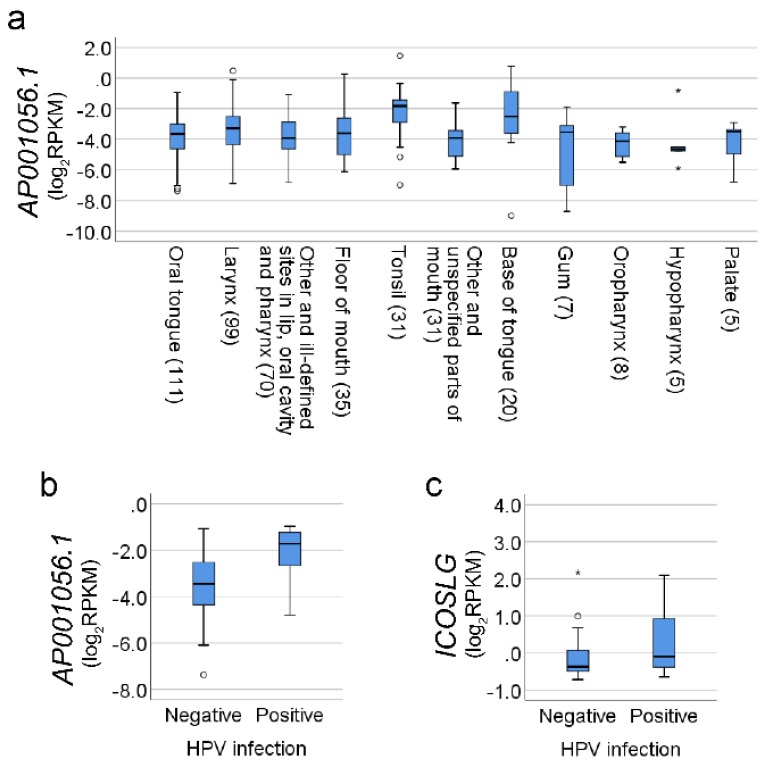
Gene expression related to tumor subsite and human papillomavirus (HPV) status. (**a**) Boxplots showing levels of *AP001056.1* in different tumor subsites. Values in brackets show number of patients. (**b**,**c**) Boxplots showing levels of *AP001056.1* and *ICOSLG* in HPV-positive and HPV-negative SCCHN tumors. Small circles indicate outliers and asterisks, (*) extreme outliers.

**Figure 3 cancers-11-00347-f003:**
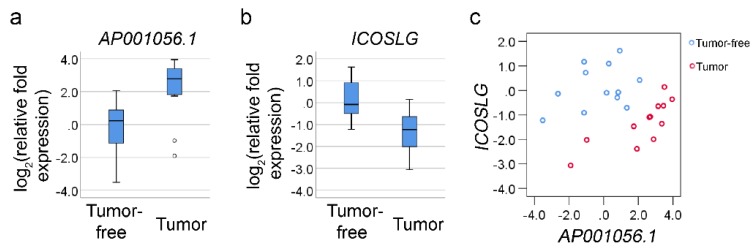
Quantification of *AP001056.1* and *ICOSLG* levels by RT-qPCR. (**a**,**b**) Boxplots showing log transformed relative fold levels of *AP001056.1* and *ICOSLG* in matched tumor-free and tumor samples from 12 patients with oral tongue SCC. Small circles indicate outliers. (**c**) Scatterplot showing correlation between *AP001056.1* and *ICOSLG* mRNA in tumor-free and tumor samples.

**Table 1 cancers-11-00347-t001:** List of overall survival associated lncRNAs derived from enhancers.

Ensembl ID	Symbol	TANRIC Overall Survival Analysis, Log-Rank *p*-Value	Predicted Target	Correlation between lncRNA and the Neighboring Target	
				*p*-value	Correlation Coefficient *r*
ENSG00000234902	*AC007879.3*	0.000	*KLF7*	No or weak correlation	
ENSG00000125514	*LINC00029*	0.000	*SLC17A9*		
ENSG00000231419	*LINC00689*	0.000	*WDR60*		
ENSG00000247982	*LINC00926*	0.001	*CGNL1*		
ENSG00000237604	*AP001056.1*	0.002	*ICOSLG*	<0.001	0.504
ENSG00000259974	*LINC00261*	0.002	*FOXA2*	No or weak correlation	
ENSG00000250334	*LINC00989*	0.003	*NAA11*		
ENSG00000229228	*LINC00582*	0.007	*DISC1*		
ENSG00000245149	*RNF139-AS1*	0.010	*NDUFB9*		
ENSG00000233208	*LINC00642*	0.014	*CALM1*		
ENSG00000235126	*AC128709.2*	0.016	*BDH1*		
ENSG00000235192	*AC009495.3*	0.019	*GALNT3*		
ENSG00000225783	*MIAT*	0.025	*CRYBB1*	< 0.001	0.401
ENSG00000237819	*AC002454.1*	0.027	*CDK6*	< 0.001	0.671
ENSG00000179818	*PCBP1-AS1*	0.030	*TIA1*	< 0.001	0.537
ENSG00000236671	*PRKG1-AS1*	0.037	*DKK1*	< 0.001	0.713
ENSG00000244342	*LINC00698*	0.039	*GALNT3*	No or weak correlation	
ENSG00000227479	*AC124861.1*	0.045	*GPC1*		

**Table 2 cancers-11-00347-t002:** TANRIC survival analysis and gene expression correlations for *AP001056.1* and *ICOSLG* across 20 cancer types from the Cancer Genome Atlas (TCGA) project.

Tumor Type		*AP001056.1* and Overall Survival Log-Rank *p*-Value	*AP001056.1* and *ICOSLG*	
Abbreviation	Detail		Correlation *p*-Value	Correlation Coefficient
HNSC	Head and Neck squamous cell carcinoma	0.002	< 0.001	0.504
GBM	Glioblastoma multiforme	0.035	< 0.001	0.571
KIRC	Kidney renal clear cell carcinoma	0.094	No or weak correlation	
THCA	Thyroid carcinoma	0.221	No or weak correlation	
LUSC	Lung squamous cell carcinoma	0.254	< 0.001	0.494
LUAD	Lung adenocarcinoma	0.262	< 0.001	0.510
BLCA	Bladder Urothelial Carcinoma	0.345	< 0.001	0.559
OV	Ovarian serous cystadenocarcinoma	0.352	No or weak correlation	
PRAD	Prostate adenocarcinoma	0.456	< 0.001	0.481
KIRP	Kidney renal papillary cell carcinoma	0.511	No or weak correlation	
LGG	Brain Lower Grade Glioma	0.558	< 0.001	0.462
STAD	Stomach adenocarcinoma	0.588	0.002	0.455
LIHC	Liver hepatocellular carcinoma	0.805	No or weak correlation	
BRCA	Breast invasive carcinoma	0.808	No or weak correlation	
CESC	Cervical squamous cell carcinoma and endocervical adenocarcinoma	0.982	< 0.001	0.412
COAD	Colon adenocarcinoma	Value not shown	No or weak correlation	
KICH	Kidney chromophobe			
READ	Rectum adenocarcinoma			
SKCM	Skin cutaneous melanoma			
UCEC	Uterine corpus endometrioid carcinoma			

**Table 3 cancers-11-00347-t003:** List of immune genes associated with *AP001056.1* expression (*r* ≥ 0.600, *p* < 0.001).

Gene Symbol	Spearman Correlation Coefficient *r*	Gene Symbol	Spearman Correlation Coefficient *r*	Gene Symbol	Spearman Correlation Coefficient *r*
LRMP	0.702	IRF8	0.646	ENPP2	0.627
NLRC3	0.696	ITK	0.645	RHOH	0.625
DOCK2	0.691	NFAM1	0.644	FCGR2C	0.625
PIK3CG	0.687	ATP8B4	0.644	CR1	0.625
PTPRC	0.683	FGR	0.643	LCP2	0.623
CD180	0.683	ALOX5	0.643	TLR10	0.618
BTK	0.683	NCF4	0.641	FLT3	0.618
ARHGAP9	0.683	CYSLTR1	0.641	CTSS	0.617
IKZF1	0.677	SPN	0.640	SELL	0.614
IL16	0.674	HVCN1	0.639	CD48	0.614
BIN2	0.670	CCR2	0.639	SELPLG	0.613
APBB1IP	0.670	BTLA	0.639	LY9	0.613
SLAMF1	0.668	TLR7	0.638	CXCR4	0.613
NCKAP1L	0.666	CD5	0.637	CORO1A	0.613
CD53	0.664	PRKCB	0.636	CD96	0.613
TLR1	0.663	PREX1	0.635	ATP8A1	0.612
PLCL2	0.662	CD79B	0.635	ITGAX	0.611
JAK3	0.659	TNFRSF1B	0.634	CYBB	0.611
TNFSF8	0.658	DOCK8	0.634	CD27	0.610
VAV1	0.657	SLAMF6	0.633	GAPT	0.609
IKZF3	0.656	ITGAM	0.633	CD4	0.609
NAIP	0.655	GPR174	0.633	GPR65	0.608
DOK3	0.655	CD84	0.633	P2RX1	0.606
LILRB1	0.654	PRAM1	0.632	DOCK10	0.606
ZAP70	0.652	IL21R	0.632	IRF4	0.603
WAS	0.651	BCL2	0.631	CIITA	0.603
TBC1D10C	0.649	LAT2	0.630	IL2RG	0.602
LTA	0.649	CCR4	0.628	AMICA1	0.602
CECR1	0.648	LILRA1	0.627	CD3G	0.600
CD28	0.648	LAX1	0.627		
ITGAL	0.646	LAT	0.627

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
