# Peer review of "AP001056.1, A Prognosis-Related Enhancer RNA in Squamous Cell Carcinoma of the Head and Neck"

_cancers, 2019, doi:10.3390/cancers11030347_

Reviewer 1 Report

This is an interesting and extremely well-written manuscript describing the first study of eRNA in HNSCC  this reviewer is aware of. the measurement of lncRNA is often difficult due to the fleeting level of them and overlap in sequence. I would really like to see the melt-curves for the cybergreen-obtained gene expression values. Also, please comment on how the RNAs were harvested (i.e., Trizol, column, etc.) as this is also a technically challenging aspect of these sort of experiments.  The observed specificity in HNSCC and GBM is very intriguing; it would be nice to see a few sentences in the discussion about how eRNA's might be exploited therapeutically. 

Author Response

Response to Reviewer #1

Yes, we agree that measurement of lncRNA is often difficult. By melting curve analysis (shown below) and agarose gel electrophoresis, we confirmed that our results were correct. However, as patient samples had been consecutively collected and included in our several previous studies with different objectives, RNA extraction has been performed using different methods (Trizol or column-based kits), which could by itself affect RNA levels. Nevertheless, as paired tumor-free and tumor samples were always extracted using the same method, we believe that the comparison between paired tumor-free and tumor samples in our study is reliable.

Thanks for the suggestion! We also think the specificity of AP001056.1 in HNSCC and GBM makes it an attractive therapeutic target. As AP001056.1 is shown to be an immune-related eRNA, novel approaches that could increase AP001056.1 expression might help to induce protective immunity for effective tumor therapy. A brief discussion on the therapeutic indication of AP001056.1 is now added in the Discussion section, line 162.

Response to the reviewer's comments is shown in the attached file.

Reviewer 2 Report

This is a remarkable study evaluating the biological role of a subclass long non-coding RNA (lncRNA), termed enhancer RNA (eRNA). In particular, the Authors investigated for the first time in Head and Neck squamous cell carcinoma (HNSCC) the prognostic role of AP001056.1, an eRNA located within the tissue-specific enhancer of ICOSLG, an immune checkpoint.

The techniques utilized were appropriate and described with plenty details. This is a well-designed study with rigorous methods. The discussion is well-balanced, and the statements are supported by the data. The study is on a timely subject in view of the potential role of lncRNAs as new biomarkers for HNSCC.

I suggest adding some considerations regarding the specific HNSCC subtype considered for the validation phase of this study (i.e. squamous cell carcinoma of the oral tongue). In particular, I suggest adding some considerations related to the prognosis of squamous cell carcinoma of the oral tongue, that is one of the most prevalent tumours of the head and neck region [1].

[1] Mascitti M, et al. American Joint Committee on Cancer staging system 7th edition versus 8th edition: any improvement for patients with squamous cell carcinoma of the tongue? Oral Surg Oral Med Oral Pathol Oral Radiol. 2018 Nov;126(5):415-23.

Author Response

Thanks for the suggestions! According to TCGA clinical data, we have compared gene expression levels between different HNSCC subtypes. To further compare gene expression between tumor and tumor-free controls, we quantified AP001056.1 RNA in paired tumor and tumor-free samples from 12 patients with oral tongue cancer, the most common tumor subtype (the mentioned reference is now added in line 120). We also wanted to investigate the prognostic impact of AP001056.1 on oral tongue cancer, however, the included patient group was too small for such an analysis.